# Development and Validation of a Screening Equation for Obstructive Sleep Apnea

**DOI:** 10.3390/diagnostics15040427

**Published:** 2025-02-10

**Authors:** Antonio Fabozzi, Alessia Steffanina, Matteo Bonini, Paolo Palange

**Affiliations:** Pulmonology Unit, Department of Public Health and Infectious Diseases, Policlinico Umberto I, “Sapienza” University of Rome, 00185 Rome, Italy; alesteffanina@yahoo.it (A.S.); matteo.bonini@uniroma1.it (M.B.); paolo.palange@uniroma1.it (P.P.)

**Keywords:** obstructive sleep apnea, sleep apnea syndromes, disorders of excessive somnolence, screening, polysomnography, obesity

## Abstract

**Background:** The high prevalence of obstructive sleep apnea (OSA), about 30% of people worldwide over 30 years old, underscores the crucial need for early screening. This study aimed to identify key predictive factors for OSA; use these factors to develop a screening equation for a population at high risk for OSA; and prospectively validate this equation’s application. **Methods:** The study included two phases: a retrospective phase examining anthropometric data, the Epworth sleepiness scale (ESS), and the home sleep apnea test (HSAT) from 200 patients referred to the Respiratory Sleep Disorder Center at Policlinico Umberto I, Rome, Italy (January 2020–January 2023) to create a predictive equation for OSA using multivariate analysis (with the most predictive data according to scientific literature). A prospective phase testing this equation on 53 patients from May 2023 to September 2024. **Results:** In the retrospective phase, the most predictive variables for the apnea–hypopnea index (AHI) identified were neck circumference (NC) and the Epworth sleepiness scale (ESS). The predictive equation derived from the multivariate analysis was as follows: AHIp = [−70.498 + (2.196 × NC) + (0.809 × ESS)]. In the prospective phase of the study, we compared the AHI predicted by the equation (AHIp) with the AHI measured via the HSAT (AHIm) in 53 patients recruited. The results showed that AHIp had a sensitivity of 95%, a specificity of 28%, a positive predictive value (PPV) of 46%, and a negative predictive value (NPV) of 90%. **Conclusions:** This study identified NC and ESS as key predictors of OSA, forming a predictive equation. This equation, showing high sensitivity and high NPV, may be useful as a screening method to rule out OSA.

## 1. Introduction

### 1.1. The Burden of Obstructive Sleep Apnea

Obstructive sleep apnea (OSA) is a respiratory disorder characterized by recurrent episodes of partial or complete obstruction of the upper airway during sleep, leading to intermittent hypoxemia and sleep fragmentation [1,2,3]. OSA represents a risk factor for the most prevalent diseases worldwide, such as arterial hypertension, cardiovascular diseases, type 2 diabetes, and neurocognitive disorders [4,5,6]. Moreover, OSA significantly impacts patients’ quality of life and public health [7,8]. In fact, OSA causes decreased productivity, an increased rate of road accidents due to excessive daytime sleepiness (EDS), and a high burden on the healthcare economy [9,10,11]. Given the high prevalence of OSA, its consequences on human health, and its impact on the social and economic sphere, there is a critical necessity for early detection. Increased public awareness and easy-to-apply screening programs in the general population are needed to achieve this objective.

### 1.2. Epidemiology and Risk Factors

Recent systematic reviews estimate a global prevalence of OSA in individuals aged 30–69 years in a range between 17% and 30%, considering an apnea–hypopnea index (AHI) ≥ 15 events/h as the diagnostic threshold [12,13]. Among the older age population, OSA prevalence rises significantly to 36–50% [14,15]. In fact, oropharyngeal resistance increases, and upper airway dimensions decrease with old age [16,17,18]. The association between obesity and OSA is very strong [16,19,20]. In fact, both the prevalence and severity of OSA increase in parallel with rising levels of BMI [21,22]. However, BMI is not the best biomarker to identify the obese population at high risk for OSA. Actually, neck circumference (NC) and waist-to-hip ratio better represent the amount of visceral fat, known to be strongly associated with the severity of OSA [23,24]. For example, a recent study shows that an NC ≥ 38 cm had a sensitivity of 58% and specificity of 79% in predicting the presence of OSA [25]. The association between obesity and OSA is also confirmed by the decreased AHI after weight loss, whether by bariatric surgery or intensive lifestyle intervention [26,27].

### 1.3. The State of Art of OSA Screening

The screening for OSA is conducted through questionnaires about symptoms, such as the ESS (Epworth Sleepiness Scale) and the Berlin questionnaire. The ESS estimates the risk of OSA through the detection of excessive daytime sleepiness (EDS) [28], which is, however, a symptom present in only 40% of OSA patients [29]. In addition, EDS can also be caused by neurodegenerative or neuromuscular diseases, sleep deprivation, or abuse of hypnotic-sedating drugs [30]. For these reasons, ESS showed a low pooled sensitivity but a high pooled specificity to detect OSA [31,32]. Another commonly used symptom-based questionnaire is the Berlin questionnaire, which investigates the presence of snoring/choking during sleep [33]. The main limitation of the Berlin questionnaire concerns women, as the symptomatology they often complain about consists of insomnia, anxiety, depression, nightmares, and palpitations, which are not included in the present questionnaire [34]. Other commonly used questionnaires such as the STOP-Bang questionnaire and the multivariable apnea prediction index include, in addition to symptoms, several risk factors for OSA. The STOP-Bang questionnaire includes, for example, obesity, male sex, increased neck circumference, and high blood pressure [35]. A limitation of the STOP-Bang questionnaire is its use of fixed cut-offs of anthropometric parameters such as BMI and NC, which may not be appropriate for different geographical areas [36]. Furthermore, it considers that all variables contribute equally to the total score of the questionnaire, without calculating the specific weight of each of the risk factors [37]. 

To date, there is not enough evidence on the accuracy of these screening questionnaires for detecting OSA [38]. In a recent meta-analysis, considering an AHI ≥ 15 events/h as the threshold, both the Berlin questionnaire and the STOP-Bang questionnaire showed a high pooled sensitivity and lower pooled specificity to detect moderate OSA [31]. 

Moreover, there is ongoing debate about the necessity and feasibility of OSA screening in the general population. In 2022, the US Preventive Services Task Force stated that the evidence remains insufficient to conclude that the benefits of routine OSA screening outweigh the potential harms in asymptomatic adults or those with unrecognized OSA symptoms [39]. This uncertainty has resulted in a diagnostic delay in the detection of moderate to severe OSA. This gap is particularly worrying considering that almost 80% of USA patients with moderate or severe OSA are undetected in primary care, causing a significant impact on the healthcare economy and the rate of road accidents [40]. Undiagnosed moderate–severe OSA burdens the healthcare economy significantly due to comorbid conditions, lost productivity, and higher rates of EDS-related accidents [41]. These missed diagnoses also delay treatment, worsening cardiovascular and metabolic outcomes [42].

### 1.4. Aim of the Study

The primary aim of our study was to identify key anthropometric and clinical factors suggestive of OSA in a real-life population referred to our Sleep Disorder Center for sleep-related symptoms. Specifically, we wanted to determine the most predictive variables for the apnea–hypopnea index (AHI). The secondary objective was to develop a simple predictive equation for estimating the AHI. Thus, we aimed to prospectively validate this equation in a separate cohort, evaluating its accuracy, sensitivity, and specificity, as a potential screening tool in a population at high risk for OSA.

## 2. Materials and Methods

### 2.1. First Phase

In the first phase, we retrospectively analyzed the medical files of 200 consecutive patients referred to our Respiratory Sleep Disorder Center of Pneumology, Policlinico Umberto I, Rome, Italy, between January 2020 and January 2023. Patients were recruited according to their referral for sleep-disordered breathing. The parameters routinely recorded in our clinical practice and included in our study were: age, sex, BMI, NC, ESS, comorbidities, and smoking habits. All 200 patients met the inclusion and exclusion criteria. Inclusion criteria included: age ≥ 18 years old and the execution of a home sleep apnea test (HSAT) on the first visit. Exclusion criteria were a diagnosis of other non-respiratory sleep disorders or neuromuscular diseases. The HSAT was conducted under the guidelines of the American Academy of Sleep Medicine (AASM) [43] using the SOMNO touch RESP device (SOMNOmedics Italia, Ora, Bolzano, Italy) and analyzed via the DOMINO light software (V1.5.0.12) (SOMNOmedics Italia, Ora, Bolzano, Italy) [44]. Apnea was defined as a decreased peak signal excursion of ≥90% from the pre-event baseline for ≥10 s using a nasal airflow sensor [43]. Hypopnea was defined as a decreased peak signal excursion of ≥30% compared to the pre-event baseline for ≥10 s, compared with ≥3% arterial oxygen desaturation [43]. We collected the following data from medical records: age, body mass index (BMI), OSA-related symptoms, ESS score, NC, the presence of comorbidities (type 2 diabetes, arterial hypertension, heart failure, dyslipidemia, gastroesophageal reflux disease), and the AHI.

### 2.2. Second Phase

In the second phase, we prospectively analyzed the medical files of 53 patients referred between May 2023 and September 2024 for their first medical visit to our center. The sample size was calculated considering an estimated prevalence of moderate to severe OSA (AHI ≥ 15 events/h) of 60%, as in our retrospective phase. We assumed for the predictive equation an expected sensitivity of 90%, with a 10% margin of error and a 95% confidence level. The minimum required sample size was 50 patients. We increased the sample size to 53 patients, considering an eventual data loss or incomplete follow-up. The HSAT was performed using identical procedures and definitions as in the first phase.

### 2.3. Statistical Analysis

All the statistical analyses were conducted using Jamovi software (V2.3.28.0). Data were summarized as means and standard deviations for continuous variables and frequencies for categorical variables. We used a one-way analysis of variance (ANOVA) to compare the means of continuous variables between groups, while we used the chi-square test for categorical variables. We used the Pearson correlation coefficient and univariate linear regression analysis for the calculation of associations between the AHI and the potential predictors (BMI, age, sex, smoking habits, NC, and ESS). We performed a multivariate linear regression analysis for AHI, with the objective of identifying the best clinical predictors of AHI. We singularly tested the interactions between the main variables studied (age, sex, BMI, NC, ESS). We tested the multivariate linear regression assumptions: the linearity (through the univariate linear regression analysis and Pearson correlation), the homoscedasticity (through the Breusch–Pagan test), the residual normality (through the Shapiro–Wilk test) and the multicollinearity [trough the variance inflation factor (VIF)]. We then used the following formula to calculate the predictive equation: “AHIp = β_0_ + (β_1_ × NC) + (β_2_ × ESS)”, where β_0_ corresponds to the intercept, while β_1_ and β_2_ are the coefficients obtained from the linear regression for the AHI for the two most predictive factors (NC and ESS). The Bland–Altman test was applied to assess the concordance between the AHI predicted by our equation (AHIp) and the AHI measured by the HSAT (AHIm). The mean difference (bias) and limits of agreement (LoA) were calculated. A scatter plot of the differences compared to the means was generated to visualize the concordance. The Pearson correlation coefficient (r) was also calculated to assess the linear relationship between AHIp and AHIm. The level of statistical significance was set at *p* < 0.05.

## 3. Results

### 3.1. First Phase

A total of 200 patients (130 men and 70 women) were retrospectively recruited for our study (Table 1). Class I obesity (mean BMI: 34 ± 9 kg/m^2^) and large NC (mean: 42 ± 4 cm) were highly prevalent in our cohort. The mean ESS score was 9 ± 5, which is at the lower limits of the positivity threshold. The most frequently reported OSA-related symptom was snoring (92%), while the most associated comorbidities were hypertension, gastroesophageal reflux disease (GERD), and type 2 diabetes mellitus (T2DM). Of the cohort, 120 patients (60%) presented moderate or severe OSA (AHI ≥ 15 events per hour). NC and ESS were strongly correlated with AHI (r = 0.6, *p* < 0.001 and r = 0.54, *p* < 0.001, respectively) (Figure 1).

The univariate linear regression analysis using for AHI as dependent variable, showed that NC, ESS, BMI and age were the most associated variables with AHI (Table 2). However, when we performed the multivariate linear regression analysis using AHI as the dependent variable, we identified only NC and ESS as the most predictive variables for AHI (Table 3). We singularly tested the interactions between NC, BMI, ESS, sex, and age: none of them was statistically significant, suggesting that NC and ESS were independently predictive of AHI without relevant modulation by other variables. The multivariate linear regression assumptions were satisfied. The distribution of residuals was normal (Shapiro–Wilk test, *p* = 0.12), and homoscedasticity was verified (Breusch–Pagan test, *p* = 0.21). The VIF values ranged from 1.2 to 2.5, showing that there was no significant multicollinearity. Based on these findings, we developed the following predictive formula: *AHIp = [−70.498 + (2.196 × NC) + (0.809 × ESS)]*, where AHIp represents the predicted AHI, while NC and ESS represent the two strongest predictors for the AHI indicated by the regression analysis.

### 3.2. Second Phase

In the second phase, we prospectively analyzed the data of 53 patients (41 men and 12 women) referred to our center from May 2023 to September 2024 (Table 4). The mean BMI (34 ± 9 kg/m^2^) and the mean NC (43 ± 4 cm) were similar to the first phase patients. The self-administered ESS was positive (mean ESS: 9 ± 5 points). As in the retrospective phase, the most frequent comorbidities in this second phase were hypertension, GERD, and T2DM (75%, 58%, and 47%, respectively). In total, 75% of patients presented moderate-to-severe OSA (mean AHIm: 27 ± 14 events per hour).

At this point, we used the predictive equation calculated in the first phase in order to determine the AHIp of our 53 patients (mean AHIp: 28 ± 16 events/hour). The concordance between AHIm and AHIp was assessed through the Bland–Altman test (Figure 2). The mean difference (bias) between AHIm and AHIp was −1.13 events/hour, which was not statistically significant (*p* = 0.48), indicating that the AHIp values were very close to the corresponding values of AHIm. Moreover, most of the values fell within the range between the lower LoA (−23 events/hour) and the upper LoA (+22 events/hour). AHIm and AHIp were strongly correlated, according to the Pearson correlation test (r = 0.89, *p* < 0.001) (Figure 3).

To further evaluate the predictive accuracy of our equation, we stratified the Bland–Altman analysis by AHIm values (AHIm < 20 events/hour, 20 ≤ AHIm ≤ 40 events/hour, AHIm > 40 events/hour) (Table 5). The examples in which the AHIm was <20 events per hour or >40 events per hour were the ones in which the AHIm and AHIp correlated most strongly (R^2^ of 0.91 and 0.95, respectively). However, an AHIp < 20 events/hour had a positive mean bias (+6.25 events/hour), indicating a slight tendency to overestimate true values. In contrast, an AHIp > 40 events/hour showed a negative mean difference (−22 events/hour), underestimating the true values.

With the Bland–Altman analysis subdivided by AHI levels, we found an AHIp < 18 events/hour as the threshold for discriminating the risk of significant apneas measured by the HSAT (AHIm ≥ 20 events/hour). In fact, using an AHIp of 18 events/hour as the threshold, no false negatives were detected: in other words, there were no cases where AHIp was <18 events/hour while AHIm was ≥20 events/hour.

Finally, our predictive equation for AHI showed a sensitivity of 95%, a specificity of 28%, a positive predictive value (PPV) of 46%, and a negative predictive value (NPV) of 90%.

## 4. Discussion

### 4.1. The Impact of Neck Circumference and Epworth Sleepiness Scale

Our study, conducted on a population of 200 patients referred to our center, aimed to identify predictive variables for OSA in a high-risk population. We used the AHI as the dependent variable, as it is still the most used index to classify patients with obstructive sleep apnea. In the univariate linear regression analysis NC, ESS, BMI, and age were significantly associated with the AHI. However, the multivariate model showed that the effects of BMI and age on the AHI may be explained by NC and ESS, identified as the most predictive variables for the AHI in our study. These results are in agreement with the current literature. In fact, NC is currently recognized as one of the strongest predictors of OSA, helping to identify individuals at high risk of upper airway collapse. An increased neck circumference frequently indicates a significant neck fat deposition. Neck fat deposition tends to reduce the cross-sectional caliber of the upper airways and increases external pressure on them, leading to more collapsibility of the upper airways during sleep [45,46]. NC is an easy parameter to measure and does not require any special instrumentation or calculation skills [47]. Furthermore, the application of the NC within our predictive equation allows us to avoid the use of fixed cut-offs that may vary according to different geographical areas [48,49,50]. Precisely for this reason, there is currently no consensus on a threshold value of NC that mandates OSA screening. While the STOP-Bang questionnaire sets this threshold at 43 cm for men and 41 cm for women [51], more recent studies identify lower thresholds (36.5 cm for women and 41 cm for men) [52]. On the other hand, the NoSAS score (Neck, Obesity, Snoring, Age, Sex) score identified a cut-off of 40 cm, without any gender differences [53]. Similarly, ESS is considered an important tool to identify the most symptomatic population to undergo an HSAT [28]. The ESS identifies the “excessively sleepy” OSA phenotype, which is at increased risk of cardiovascular events and other relevant comorbidities [54]. The advantages of the ESS include its brevity, accessibility, and feasibility. Furthermore, it is easy to understand and can be completed in a self-administered format by the patient [55]. However, there is no consensus on which cut-off should be used for test positivity [56]. Some studies propose a cut-off value of 8 points [57], while others suggest 10 or 11 points [31]. Furthermore, its accuracy varies depending on whether it is self-administered or administered by healthcare professionals. Finally, ESS presents considerable variation according to gender, age, and comorbidities [58]. To address these limitations, our approach was to avoid the use of a cut-off for OSA screening. Instead, we propose a predictive equation that combines NC and ESS, both with a relative specific weight, to provide a personalized screening tool for OSA.

### 4.2. The Predictive Equation

Our predictive equation represents a promising tool for screening OSA in high-risk populations. By combining NC and ESS, we developed a predictive equation to estimate the AHI (AHIp) measured by the HSAT (AHIm). The high sensitivity (95%) and the high negative predictive value (90%) of our equation make it particularly effective as a screening tool. In fact, an AHIp below 18 events per hour confidently excludes moderate–severe OSA measured by an HSAT. This method may reduce the gap in undetected patients, as up to 80% of moderate–severe OSA cases remain undiagnosed, highlighting the need for effective screening tools [59]. Actually, several predictive models have been described in the literature. For example, a study by Amra B. et al. showed that a combination of Mallampati score > 2, age > 51 years, and NC > 36 cm estimated an AHI ≥ 15 events/hour in 94% of cases. However, no specific equation was developed to predict the AHI [60]. Similarly, the DES–OSA score uses morphological and anthropometric parameters such as Mallampati score, BMI, sex, and age to predict AHI with a sensitivity of 82% [61]. Other predictive models for AHI, such as the formula by Sahin M. et al., include variables like peripheral oxygen saturation and tonsil size [62]. However, these predictive models for the AHI do not include symptom-related variables, unlike our equation, which includes ESS to account for excessive daytime sleepiness. Moreover, NC is the only morphological parameter to be measured in our equation, simplifying its application in large-scale settings.

### 4.3. Future Applications

Patients at high risk of OSA often first interface with the general practitioner. In some geographical contexts, resources such as HSATs are limited, so having an initial screening tool such as ours available, which is feasible and takes only a few minutes, is crucial. Moreover, it permits the rapid exclusion of moderate–severe cases of OSA. Our equation can also be applied in telemedicine or remote monitoring. Our equation can become part of public health screening programs, educating the general population about the right awareness of risk factors and symptoms related to OSA.

### 4.4. Study Strengths and Limitations

Our equation is easily applicable in numerous clinical contexts, from primary care to sleep clinics. It can be drawn up for large-scale screening or in settings with limited resources (low possibility of performing the HSAT). It also allows the identification of different phenotypes of OSA, such as the non-obese “excessively sleepy”. It also avoids the use of rigid cut-offs or a scoring system. Finally, it takes only a few minutes, as all that is needed is to complete a questionnaire and possess a dressmaker’s tape measure. 

However, possible limits of the present study lie in its retrospective, observational, and monocentric nature. The low sensitivity of the equation raises the risk of false positives, resulting in performing unnecessary HSATs. Another limitation is that the anthropometric measures, such as NC, were obtained from a cohort of Italian patients referred to our sleep disorders center. It is well known that NC can vary significantly between different geographic areas and ethnic groups. For this reason, this geographical variability of NC could affect the generalizability of our equation to populations outside Italy. Another limitation of our study is the use of the HSAT instead of polysomnography to diagnose OSA. In fact, current guidelines confirm laboratory polysomnography as the diagnostic gold standard for OSA. We chose the HSAT for its cost-effectiveness and practicality in a real-world clinical setting. However, the HSAT may have underestimated both the number of patients with OSA and the severity of the disease. This potential underestimation must be considered in the interpretation of our findings.

## 5. Conclusions

In conclusion, our study developed and validated an effective predictive equation to estimate the AHI in high-risk populations for OSA. By combining a morphological parameter (NC) and a symptom-based parameter (ESS), the equation showed high sensitivity and high negative predictive value, effectively ruling out all cases of moderate–severe OSA. Specifically, an AHIp below 18 reliably excludes an AHI ≥ 20 events/hour measured by the HSAT. Further studies are needed to optimize the equation and improve its specificity. Further larger studies are required to validate its application to primary care and the general population.

## Figures and Tables

**Figure 1 diagnostics-15-00427-f001:**
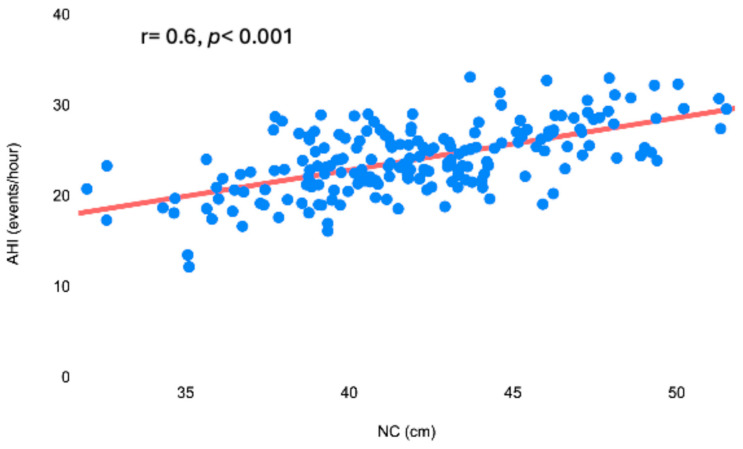
Significant positive correlation between AHI and NC. AHI, apnea–hypopnea index; NC, neck circumference.

**Figure 2 diagnostics-15-00427-f002:**
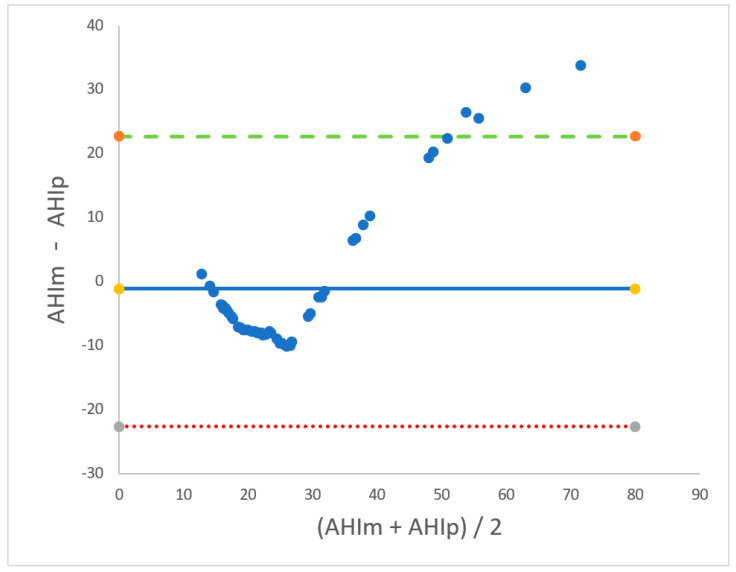
Graphical evaluation of the concordance between AHIm and AHIp through the Bland–Altman plot. The X-axis represents the mean of AHIm plus AHIp, while the Y-axis represents the difference between AHIm and AHIp. The blue solid line represents the bias value (mean difference between AHIm and AHIp). The green dashed line represents the upper limit of agreement (LoA), and the red dashed line represents the lower LoA.

**Figure 3 diagnostics-15-00427-f003:**
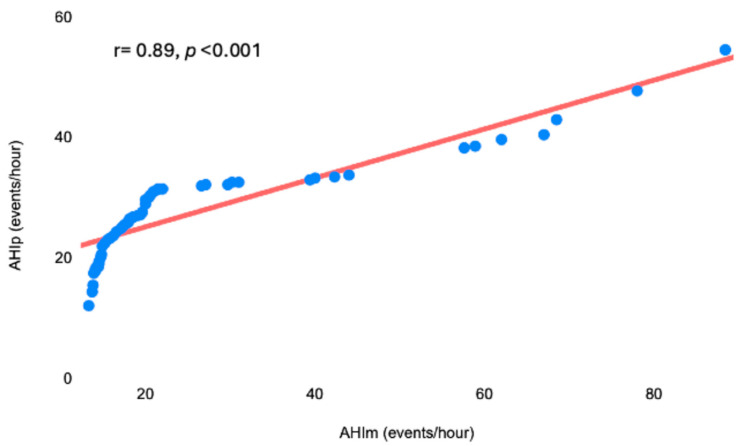
Significant linearly positive correlation between AHIm and AHIp. AHIm, measured apnea–hypopnea index; AHIp, predicted apnea–hypopnea index.

**Table 1 diagnostics-15-00427-t001:** Clinical and demographic characteristics of the retrospective cohort.

	Number of Patients: 200
Men/Women, *n* (%)	130 (65)/70 (35)
Age, years	54 (13)
Smoking habits, *n* (%)	Nonsmokers: 92 (46)
	Smokers: 74 (37)
	Former smokers: 34 (17)
T2DM, *n* (%)	80 (40)
Hypertension, *n* (%)	110 (55)
Dyslipidemia, *n* (%)	24 (12)
GERD, *n* (%)	86 (43)
Heart failure, *n* (%)	6 (3)
BMI, kg/m^2^	34 ± 9
NC, cm	42 ± 4
Snoring, *n* (%)	190 (95)
Reported apneas, *n* (%)	50 (25)
ESS	9 ± 5
AHI, events per hour	25 ± 14

Values are means ± standard deviation. T2DM, type 2 diabetes mellitus; GERD, gastroesophageal reflux disease; BMI, body mass index; NC, neck circumference; ESS, Epworth sleepiness scale; AHI, apnea–hypopnea index.

**Table 2 diagnostics-15-00427-t002:** Univariate linear regression analysis for the AHI (dependent variable).

Independent Variable	β Coefficient	OR (95% CI)	*p* Value
NC	2.312	1.67 (1.55–1.81)	<0.001
ESS	0.925	2.24 (2.00–2.50)	<0.001
BMI	1.784	1.53 (1.34–1.72)	0.002
Age	0.298	1.03 (1.01–1.05)	0.015
Sex	1.364	1.18 (0.97–1.42)	0.087
Smoking habits	0.542	1.10 (0.92–1.32)	0.213

AHI, apnea–hypopnea index; OR, odds ratio; NC, neck circumference; ESS, Epworth sleepiness scale, BMI, body mass index.

**Table 3 diagnostics-15-00427-t003:** Multivariate linear regression analysis for the AHI (dependent variable).

Independent Variable	β Coefficient	OR (95% CI)	*p* Value
Intercept	−70.498	-	
NC	2.196	1.47 (1.34–1.61)	<0.001
ESS	0.809	2.06 (1.81–2.34)	<0.001
Sex	1.253	1.15 (0.92–1.43)	0.217
Age	0.298	1.01 (0.86–1.13)	0.138
BMI	0.874	1.12 (0.97–1.29)	0.101
Smoking habits	0.462	1.08 (0.89–1.30)	0.345

AHI, apnea–hypopnea index; OR, odds ratio; NC, neck circumference; ESS, Epworth sleepiness scale, BMI, body mass index.

**Table 4 diagnostics-15-00427-t004:** Patients’ clinical and demographic features.

	Number of Patients: 53
Men/Women, *n* (%)	41 (77)/12 (23)
Age, years	62 ± 15
	Nonsmokers: 19 (36)
Smoking habits, *n* (%)	Smokers: 21 (40)
	Former smokers: 13 (24)
T2DM, *n* (%)	25 (47)
Hypertension, *n* (%)	40 (75)
Dyslipidemia, *n* (%)	18 (34)
GERD, *n* (%)	31 (58)
BMI, kg/m^2^	32 ± 8
NC, cm	43 ± 4
ESS	9 ± 5
AHIm, events per hour	27 ± 14
AHIp, events per hour	28 ± 16

Values are means ± standard deviation. T2DM, type 2 diabetes mellitus; GERD, gastroesophageal reflux disease; BMI, body mass index; NC, neck circumference; ESS, Epworth sleepiness scale; AHIm, measured apnea–hypopnea index; AHIp, predicted apnea–hypopnea index.

**Table 5 diagnostics-15-00427-t005:** Stratified Bland–Altman analysis by AHIm levels.

	AHIm < 20 Events/Hour	20 ≤ AHIm ≤ 40 Events/Hour	AHIm > 40 Events/Hour
Bias, events/hour	6.25	4.85	−22
Lower LoA, events/hour	1.14	−6.9	−38.3
Upper LoA, events/hour	11.3	16.6	−5–5
R^2^	0.91	0.81	0.95

AHIm, measured apnea–hypopnea index; LoA, limit of agreement; R^2^, coefficient of determination by Pearson correlation test.

## Data Availability

The dataset used for our analysis is available upon demand to the corresponding author of this study.

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
