# Peer review of "Development and Validation of a Screening Equation for Obstructive Sleep Apnea"

_diagnostics, 2025, doi:10.3390/diagnostics15040427_

Round 1
Reviewer 1 Report
Comments and Suggestions for Authors
The authors sought to develop and validate a predictive equation of OSA (diagnosed by Home Sleep Apnea Test) from NC circumference measurement and symptoms of excessive day time sleepiness. The manuscript addresses an important gap in screening for OSA in Italian population levering on using simple metric and easily measurable symptoms.
The major weakness is insufficient documentation of statistical method used. Other concern is the generalizability of study to other geographical areas. Morphological and anthropometric measurements vary geographically. The author should discuss this as a limitation. Their result may only be applicable to Italian population. Another limitation is the use of home sleep apnea testing (HSAT) as the gold standard for diagnosing OSA. HSAT is more accessible and less expensive compared to in-laboratory polysomnography, it may underestimate the severity of OSA.
SPECIFIC COMMENTS ON OSA DIAGNOSTICS
Some minor English edits using upper cases instead of lower cases: line 36, 37
Not sure there is need to subdivide the introduction for example the section on epidemiology may not be needed. The introduction looks like a literature review
METHODS
The authors should report whether multivariate linear regression assumption were met and what is the goodness of their equation. Was collinearity check for each of the variables used? Where there missing variables in the predictor variable, given that the first part of the study is a retrospective study.
RESULTS
Table 1: The table should be properly aligned (age)
The variable scoring is reported with a continuous variable: 42 ± 4, I guess this is a mix up
Table 2: Which other predictions were used in the equation. It is good to show other predictor variables that are used in regression model
Author Response
We sincerely thank the reviewer for the constructive feedbacks provided on our manuscript. We appreciate your valuable input, which has allowed us to address important aspects of our study and further refine our analysis. Below, we provide detailed responses to each of your comments.
Comments 1: The major weakness is insufficient documentation of statistical method used. Other concern is the generalizability of study to other geographical areas. Morphological and anthropometric measurements vary geographically. The author should discuss this as a limitation. Their result may only be applicable to Italian population. Another limitation is the use of home sleep apnea testing (HSAT) as the gold standard for diagnosing OSA. HSAT is more accessible and less expensive compared to in-laboratory polysomnography, it may underestimate the severity of OSA.
Response 1: We have expanded the limitations section by adding the geographical/ethnic limitation and the one concerning the use of HSAT instead of polysomnography.
Comments 2: Some minor English edits using upper cases instead of lower cases: line 36, 37.
Response 2: The minor English edits has been issued as requested.
Comments 3: Not sure there is need to subdivide the introduction for example the section on epidemiology may not be needed. The introduction looks like a literature review.
Response 3: The epidemiology section was significantly reduced, as requested.
Comments 4: The authors should report whether multivariate linear regression assumption were met and what is the goodness of their equation. Was collinearity check for each of the variables used? Where there missing variables in the predictor variable, given that the first part of the study is a retrospective study.
Response 4: First, we included the missing variables in the multivariate analysis and we added the univariate analysis for the AHI. We have also included, in the methods and results section, the verification of multivariate linear regression assumption.
Comments 5: Table 1: The table should be properly aligned (age).
Response: Table 1 has been aligned.
Comments 6: The variable scoring is reported with a continuous variable: 42 ± 4, I guess this is a mix up.
Response 6: In our study, NC was treated as a continuous variable throughout the multivariable regression model. The reported value of 42 + 4 cm in Table 1 represents the mean ± standard deviation of NC in the studied cohort. This approach aligns with standard practices for presenting continuous variables and was consistently applied to other variables such as age, BMI, and ESS. We confirm that NC was not categorized or dichotomized in the regression analysis, allowing us to capture its full variability and relationship with the apnea-hypopnea index (AHI). This ensures that the beta coefficient reported for NC in Table 2 (β=2.196β=196) reflects the adjusted effect of NC as a continuous predictor of AHI, independent of other variables in the model. To further clarify, we have added an explanatory note to Table 1 in the revised manuscript to specify that 'Values are presented as mean ± standard deviation'.
Comments 7: Table 2: Which other predictions were used in the equation. It is good to show other predictor variables that are used in regression model.
Response 7: We have added the other variables that we considered in the multivariate analysis in Table 3
Reviewer 2 Report
Comments and Suggestions for Authors
The authors of the manuscript developed an innovative screening equation for obstructive sleep apnoea based on neck circumference (NC) and the Epworth Sleepiness Scale (ESS). The approach was to avoid the use of a cut-off for OSA screening. NC and ESS are widely accepted predictors of OSA. Their combination into a specific screening equation facilitates initial screening in primary care or home settings.
However, there are several major concerns that need to be addressed before consideration for publication:
1、In recent years, major research efforts have been made worldwide to "move OSA beyond the AHl index". As for OSA screening by performing HSAT, oxygen saturation during sleep is also an indicator that we need to pay attention to. In the manuscript, the multivariable linear regression analysis used only AHI as the dependent variable and did not take into account the lowest oxygen saturation. In addition, the authors did not describe how the two indicators (NC and ESS) were selected from the medical records. It may be appropriate to first perform a univariate linear regression analysis to screen for possible indicators, followed by a multivariate linear regression analysis.
2、As mentioned in the limitations, the low sensitivity of the equation increases the risk of false positives, resulting in unnecessary HSAT. Further optimisation of the equation is needed to improve specificity, which may be related to under-representation of the sample and collection of too few parameters, and needs to be validated in larger and more diverse samples.
3、In the methodology section, the authors also do not provide details on how to deal with potential confounders and interactions, which may affect the in-depth analysis and interpretation of the results.
4、As shown in Table 1, the clinical and demographic characteristics of the retrospective cohort had a mean BMI of 34 ± 9 kg/m2. This does not appear to be a BMI characteristic of the general population.
Author Response
We sincerely thank the reviewer for the constructive feedbacks provided on our manuscript. We appreciate your valuable input, which has allowed us to address important aspects of our study and further refine our analysis. Below, we provide detailed responses to each of your comments.
Comments 1:
Response 1: While we recognize that recent research has explored moving beyond the apnea-hypopnea index (AHI) as the sole diagnostic metric, the current international guidelines, including those from the American Academy of Sleep Medicine (AASM), continue to rely on AHI for the definition and classification of OSA. Given that our objective was to develop a screening equation aligned with the current diagnostic criteria for OSA, we chose to use AHI as the dependent variable.
We would like to clarify that the data were retrospectively collected from patients’s medical records who attended our Sleep Disorders Center. We have modified the methods section specificizing this concept. The parameters routinely recorded in our clinical practice and included in patient’s medical records are age, sex, BMI, NC, ESS, comorbidities, and smoking habits. All these variables were included in the univariate analysis and multivariate analysis. We have added the univariate analysis, as requested.
Comments 2: As mentioned in the limitations, the low sensitivity of the equation increases the risk of false positives, resulting in unnecessary HSAT. Further optimisation of the equation is needed to improve specificity, which may be related to under-representation of the sample and collection of too few parameters and needs to be validated in larger and more diverse samples.
Response 2: We agree with the reviewer that the low sensitivity of the equation increases the risk of false positives, leading to unnecessary HSAT. This limitation, as added now in the conclusions, highlights the need for further optimization of the equation to improve its specificity. We also concur that the under-representation of the sample and the limited number of collected parameters may have contributed to this limitation. Future studies will focus on validating the equation in larger and more diverse populations and exploring additional parameters to enhance its accuracy.
Comments 3: In the methodology section, the authors also do not provide details on how to deal with potential confounders and interactions, which may affect the in-depth analysis and interpretation of the results.
Response 3: We confirm that we tested the assumptions of the multivariate linear regression model, including linearity (using univariate linear regression analysis and Pearson correlation), homoscedasticity (using the Breusch-Pagan test), residual normality (using the Shapiro-Wilk test), and multicollinearity [using the Variance Inflation Factor (VIF)]. Furthermore, we singularly tested for potential interactions between key variables (sex, age, BMI, ESS and NC) but no significant interactions were identified. We have added these specifications in the methods and results section.
Comments 4: As shown in Table 1, the clinical and demographic characteristics of the retrospective cohort had a mean BMI of 34 ± 9 kg/m2. This does not appear to be a BMI characteristic of the general population.
Response 4: As noted, the cohort represents a high-risk population referred to our Sleep Disorders Center by other physicians due to clinical suspicion of OSA. This selection bias explains the elevated mean BMI compared to the general population. To address this point, we have clarified in the conclusions that future studies are needed to validate the predictive equation in larger and more diverse populations, including the general population. Additionally, we have revised the study objectives by removing the term 'general' from the phrase 'general population at high risk of OSA' to avoid potential misinterpretations. These changes have been incorporated into the revised manuscript.